# 🥷 THOUGHTTERMINATOR: Benchmarking, Calibrating, and Mitigating Overthinking in Reasoning Models

**Sophia Xiao Pu**[C*]   **Michael Saxon**[W*]   **Wenyue Hua**[M]   **William Yang Wang**[C]
[C]University of California, Santa Barbara   [W]University of Washington   [M]Microsoft
Contact: `xiao_pu@ucsb.edu, mssaxon@uw.edu`

## Abstract

Reasoning models have demonstrated impressive performance on difficult tasks that traditional language models struggle at. However, many are plagued with the problem of overthinking—generating large amounts of unnecessary tokens which don't improve accuracy on a question. We introduce approximate measures of problem-level difficulty and demonstrate that a clear relationship between problem difficulty and optimal token spend exists, and evaluate how well calibrated a variety of reasoning models are in terms of efficiently allocating the optimal token count. We find that in general, reasoning models are poorly calibrated, particularly on easy problems. To evaluate calibration on easy questions we introduce DUMB500, a dataset of extremely easy math, reasoning, code, and task problems, and jointly evaluate reasoning model on these simple examples and extremely difficult examples from existing frontier benchmarks on the same task domain. Finally, we introduce THOUGHTTERMINATOR, a training-free black box decoding technique that significantly improves reasoning model calibration.

## 1 Introduction

Investment in improving the capabilities of language models has recently turned from data- and train-time-scaling to *inference-scaling*, or training so-called *reasoning models* to expend more runtime compute generating chains of thought (Wei et al., 2022), debate (Liang et al., 2023), and self-corrections (Pan et al., 2024) in order to more robustly and correctly answer queries (Wu et al., 2024). This work appears to show a direct, positive relationship between inference spend and "reasoning task" performance (Jaech et al., 2024).

Under the inference scaling paradigm, controlling costs is critical. Unfortunately, open reasoning models such as DeepSeek r1 (DeepSeek-AI et al., 2025) and QwQ (Qwen, 2025) have demonstrated a tendency to expend unnecessary inference tokens after the answer has already could be generated, a problem referred to as *overthinking* (Chen et al., 2024).

We need to precisely define overthinking in order to mitigate it. Chen et al. (2024) define overthinking as the amount of times the model repeats the correct answer in its intermediate reasoning chain. From this definition, they used supervised fine-tuning and direct preference optimization to train reasoning models to prefer to select the shortest answer. Similar work applied knowledge distillation from non-reasoning models to blend concise answering abilities with the superior performance of reasoning models (Yang et al., 2025). However, both of these methods require retraining, a process that may be costly or have unintended consequences on performance.

Training-free methods which seek to manage overthinking include selective invocation of chain-of-thought on tasks where it has known benefit (Sprague et al., 2024), early stopping of reasoning chains using probe-based confidence on final answer tokens (Fu et al., 2024), or simply eliciting reasoning model-like behavior from non-reasoning models using continuing

---

[*]Co-first contributions.   [W&M]All work completed at the University of California, Santa Barbara.

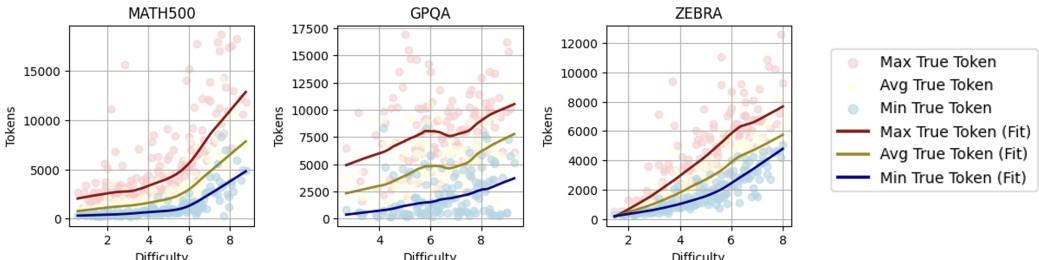

Figure 1: Question-level difficulty $D_{\mathcal{M}}(q, a)$ vs average token spend $S_{\mathcal{M}}(q)$ for Deepseek-r1 (model $\mathcal{M}$) on three reasoning datasets. Difficulty scores are scaled by 10 for readability. We observe a clear relationship between question difficulty and token spend distribution.

phrases like "wait...", which can be halted at any time (Muennighoff et al., 2025). Limitations of these methods include requiring external knowledge of task type, white-box access to the base model, or the use of non-reasoning models for precise control (Yu et al., 2025).

In this work we analyze the *token spend & difficulty calibration* of reasoning models. Starting from the supposition that more difficult problems require more thought, we first characterize this difficulty-cost relationship in a variety of open reasoning models across three reasoning datasets—MATH500 (Lightman et al., 2023), GPQA (Rein et al., 2023), and ZebraLogic (Lin et al., 2024)—allowing us to introduce a difficulty-calibrated measure of overthinking.

As these three existing datasets only allow us to assess overthinking on hard problems, we introduce DUMB500, a dataset of "easy" queries to explore overthinking on easy inputs. Finally, we introduce **THOUGHTTERMINATOR**, a *tool-based, training-free, black box decoding strategy to mitigate overthinking* using difficulty-calibrated conditioning. We show that **THOUGHTTERMINATOR** is a simple and effective way to control overthinking in reasoning models without requiring any access to gradients or training[1].

## 2 Difficulty Calibration in Reasoning Models

This work is concerned with how optimally a **reasoning model** $\mathcal{M}$, given **question pair** $(q, a)$, allocates its answer **token spend** $S_{\mathcal{M}}(a)$ with respect to the pair's **difficulty** $D(q, a)$.

Given that increased inference scale leads to higher performance across a variety of reasoning tasks, we hypothesize that the *difficulty of a question correlates with its optimal token spend.* We characterize a question's **model-specific difficulty** $D_{\mathcal{M}}(q, a)$ as a model's incorrect answer rate over $n$ samples $\hat{a}$ of that question $q$ against gold answer $a$.

$$D_{\mathcal{M}}(q, a) = p(a \neq \hat{a} \sim \mathcal{M}(q)) \approx \frac{1}{n} \sum_n \mathbb{1}(\mathcal{M}(q) \neq a) \tag{1}$$

We then compute a **model-agnostic difficulty estimate** $\bar{D}$ of $q$ as the expected difficulty $\mathbb{E}[D(q, a)]$ over a set of models $\mathbf{M}$. By assessing difficulty across a diverse collection of models, this measure provides a rough operational notion of difficulty that is both reproducible and relevant for analyzing inference efficiency of current LLMs.

$$\bar{D}(q, a) = \mathbb{E}[D(q, a)] \approx \frac{1}{|\mathbf{M}|n} \sum_{\mathcal{M}_i \in \mathbf{M}} \sum_n \mathbb{1}(\mathcal{M}_i(q) \neq a) \tag{2}$$

Each answer $\hat{a}_i$ incidentally sampled from $\mathcal{M}$ in response to question $q$ is associated with its own token spend $S_{\mathcal{M}}(\hat{a}_i)$. Is there a relationship between the difficulty of each question and the token spend that naturally occurs?

---

[1]Data and code available at github.com/SophiaPx/DUMB500/

| Model | Local overthinking $O_{\text{env}} \downarrow$ | Global overthinking $O_g \downarrow$ |
|---|---|---|
| Non-reasoning language models | | |
| Qwen2-7B-Instruct | 291 | 219 |
| Llama-3.2-1B-Instruct | 542 | 354 |
| Llama-3.2-3B-Instruct | 708 | 473 |
| Llama-3.1-8B-Instruct | 1971 | 1755 |
| gemma-2-2b-it | 148 | 152 |
| gemma-2-9b-it | 131 | 161 |
| gemma-2-27b-it | 178 | 187 |
| deepseek-llm-7b-chat | 155 | 90 |
| Reasoning language models | | |
| QwQ-32B-Preview | 2923 | 3698 |
| QwQ-32B | 13662 | 11248 |
| DeepSeek-R1-Distill-Qwen-1.5B | 5730 | 4262 |
| DeepSeek-R1-Distill-Llama-8B | 4232 | 5755 |
| DeepSeek-R1-Distill-Qwen-7B | 3881 | 4001 |

Table 1: Local and global overthinking scores (rounded to integers).

We assess the difficulty $\bar{D}$ and token spend $S_{\mathcal{M}}$ using reasoning and non-reasoning models from the DeepSeek (DeepSeek-AI et al., 2025), Qwen (Yang et al., 2024; Qwen, 2025), Gemma (Mesnard et al., 2024), and LLaMA (Dubey et al., 2024) families for all questions in MATH500 (Lightman et al., 2023), GPQA (Rein et al., 2023), and ZebraLogic (Lin et al., 2024).

Figure 1 contains scatter plots of $D_{\mathcal{M}}$ and $S_{\mathcal{M}}$ for each answer from DeepSeek-R1-7B for all three datasets. We observe that—similar to the dataset & model-wise relationships between performance and token spend documented in prior work (Muennighoff et al., 2025)—there also exists a clear relationship between question-level difficulty and average token spend.

Additionally, we note *considerable variance in the token spend between answer samples for each question.* These reasoning models exhibit considerable inconsistency in their efficiency between samples. This leads to two natural questions:

1. How **well-calibrated** are reasoning models in consistently realizing their optimal token spend per-question?
2. Is it possible to improve the calibration of reasoning models in their token spend?

## 2.1 Quantifying Overthinking

We formalize **observational overthinking**, or the failure in consistency a reasoning model has at realizing the minimum possible token spend per question.

The *observed minimum spend* of a question is the shortest reasoning chain in a set of a model's correct answers. We measure observational overthinking in terms of the difference between a model's typical token spend and this observed minimum. For questions sampled from dataset $\mathcal{D}$, the **global overthinking score** $O_g$ of a model is the mean difference between the length of each reasoning chain and the global observed minimum spend for each question.

$$O_g(\mathcal{M}) = \sum_{q \in \mathcal{D}} \left( \mathbb{E}[S(a \sim \mathcal{M}|q)] - \min_{\mathcal{M}_i \in \mathbf{M}}(S(a \sim \mathcal{M}_i|q)) \right) / |\mathcal{D}| \quad (3)$$

The **local envelope overthinking score** $O_{\text{env}}$ is the mean difference between the maximum and minimum spends for each question for each model.

$$O_{\text{env}}(\mathcal{M}) = \sum_{q \in \mathcal{D}} \left( \max[S(a \sim \mathcal{M}|q)] - \min(S(a \sim \mathcal{M}|q)) \right) / |\mathcal{D}| \quad (4)$$

Table 1 presents the calibration scores for the full set of LLama, Qwen, Gemma, and DeepSeek models we evaluated on the three datasets. These calibration scores represent

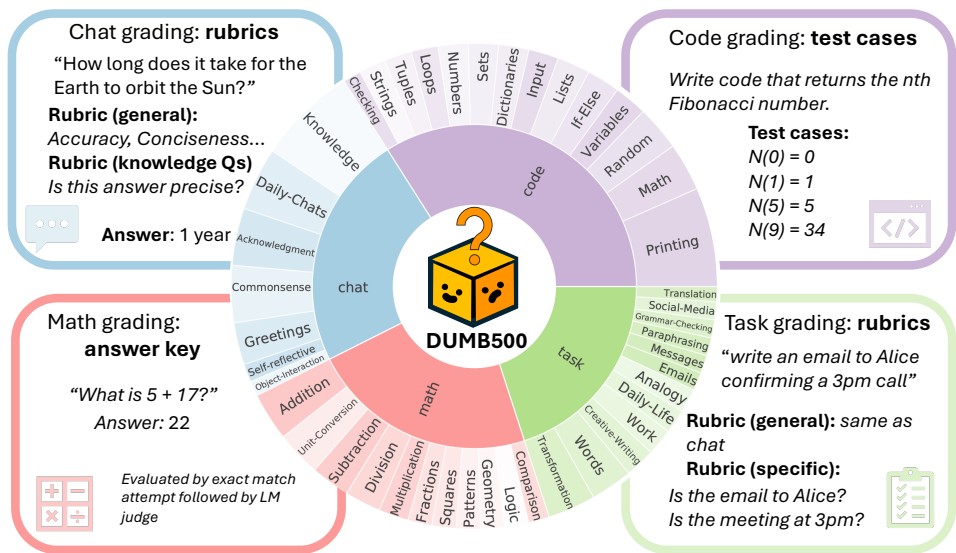

Figure 2: DUMB500 dataset composition and grading method. The dataset contains four subsets, CHAT, CODE, TASK & MATH, which are each graded with subset-specific methods. MATH are graded with traditional answer pairs. CHAT and TASK are graded using a combination of LM-judged rubrics and where appropriate, answers. CODE outputs are generated as test case coverage.

expected quantities of tokens wasted, as they are averages in excess of minimum spend values. **Lower is better.** As expected, the reasoning models with propensity to overthink have considerably higher overthinking scores than the non-reasoning models.

One weakness of our overthinking evaluation so far is that we have very few questions that have a low difficulty but high overthinking tendency. As reasoning models are most useful for challenging frontier tasks, easy question benchmarks for reasoning models don't exist. However, to get a full picture of the overthinking problem, it must also be evaluated on simple prompts. In the next section we introduce a resource, DUMB500, to do this.

## 3 Extending Overthinking Evaluation with DUMB500

While it is common knowledge that reasoning models tend to overthink on simple queries (Chen et al., 2024), no resource has been proposed to systematically evaluate this tendency on simple, straightforward questions.

The DUMB500 dataset is specifically designed to evaluate models on simple questions that humans can answer effortlessly. The goal is not to challenge models with intricate logic but rather to assess their fundamental ability to recognize simplicity and provide concise, correct responses. To the best of our knowledge, DUMB500 is the first dataset explicitly focused on extremely simple (and sometimes deliberately naive) questions. DUMB500 consists of 500 manually curated questions spanning four domains:

- **Mathematics (Math)**: Basic arithmetic, comparisons, geometric properties, and logical reasoning.

- **Conversational Interaction (Chat)**: Casual dialogue, self-reflection, common knowledge, and basic object interactions.

- **Programming & Computing (Code)**: Fundamental coding concepts, including variables, loops, conditionals, and data structures.

- **Task Execution (Task)**: Simple natural language processing tasks such as paraphrasing, translation, and basic writing.

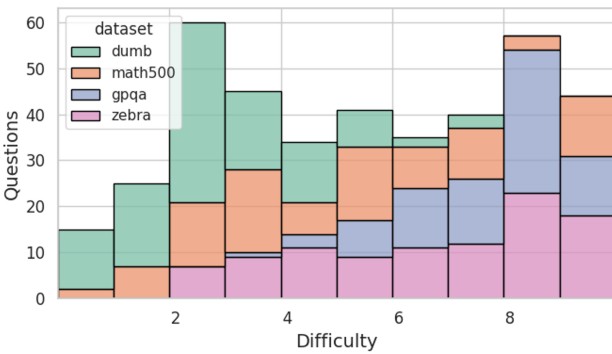

Figure 3: Joint difficulty distribution of the four datasets we evaluate in this work. Difficulty scores are scaled by 10 for readability. DUMB500 helps balance the distribution of question difficulty in our analysis, enabling a fuller assessment of reasoning model overthinking.

Each question is designed to be trivial for humans, requiring minimal cognitive effort, while still serving as a litmus test for overthinking in reasoning models. The dataset allows us to evaluate models based on two key dimensions:

- *Accuracy*: Can the model correctly answer simple questions?
- *Efficiency*: Does the model avoid unnecessary elaboration?

We manually crafted 500 simple questions across a diverse range of common knowledge, arithmetic, and practical queries. The full list of question classes with their descriptions are listed in subsection A.1. Figure 2 shows the distribution of question types in DUMB500 as well as sample questions and answers.

## 3.1 Evaluation techniques for DUMB500

In addition to the extremely simple MATH questions presented in DUMB500 which are evaluated using simple accuracy methods—identically to MATH500, GPQA, and ZebraLogic—we also introduced CHAT, CODE, and TASK questions, which require more sophisticated evaluation. They are evaluated as follows:

**CODE** questions include a set of test cases for the program described in the prompt. A python-based autograder checks that the requirements are met.

**CHAT** questions belong to one of seven subtasks (eg., greetings, acknowledgement). All chat answers are evaluated according to a set of **generic requirements**, such as *appropriateness* and *conciseness*. Depending on the subtask, **specific requirements** such as *precision* and *accuracy* are checked. When accuracy assessment is required, an answer is also provided.

**TASK** questions generally include instructions for the assistant to produce some kind of writing or answer some work-related question. In additino to using the same **generic requirements** as CHAT, TASK questions have one or more question-specific requirements which check that the implicit instructions in the prompt are followed (See Figure 2). The CHAT and TASK requirements are checked using an LM (gpt-4o-mini) as a judge[2].

## 3.2 From Dumb to Hard Questions

We evaluate the same set of models as in Table 1 on DUMB500 and analyze their accuracy and token spend across different subsets. Figure 3 depicts the distribution of questionwise difficulty scores across the MATH subset of DUMB500, MATH500, GPQA, and ZebraLogic, assessed using those models. This confirms that DUMB500-MATH fills in a gap in our analysis, adding a considerable quantity of easy questions with which to analyze overthinking.

---

[2]LM judge performance discussed in subsubsection A.2.4

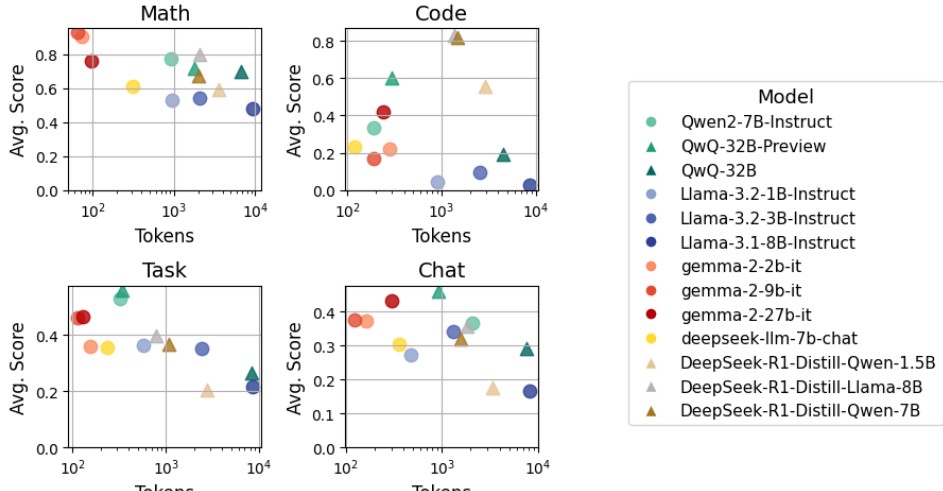

Figure 4: Relationship between average token spend $Sp$ (Tokens) and average score for the evaluated models on each subset of DUMB500.

Figure 4 shows the relationship between model-level accuracy and token spend for the tested models. As expected, on these simple math questions there is no positive relationship between token spend and accuracy, as these questions are extremely easy. For the other domains, we observe a negative correlation[3] between token spend and evaluation requirement pass rate (labeled accuracy).

## 4    THOUGHTTERMINATOR

Reasoning models often express inference scaling in natural language through tokens expressing uncertainty, like "wait..." or "let me check this..." (Muennighoff et al., 2025) Thus, overthinking often manifests as a tendency to overuse these extending expressions superfluously after the correct answer has already been found.

From this insight, we hypothesize that simple text-augmentation methods can be used to counteract this tendency, reminding the model of how long its output has been, and how soon it should come to an answer. THOUGHTTERMINATOR realizes this as a series of interrupt messages at a fixed token interval which are inserted into the autoregressive stream, alerting the model of how many tokens it has spent and how many remain.

Sometimes, these timing messages and reminders alone are sufficient to get the model to provide its answer in a concise manner. If a answer isn't provided before the end of the time limit, a terminating prompt and constrained decoding forces the model to output a final answer.

Figure 5 shows an example of a base reasoning model and one using THOUGHTTERMINA-TOR answering a question. THOUGHTTERMINATOR operates on a reasoning chain in three stages: **scheduling**, **running**, and **terminating**.

**Scheduling.**   Given an input question THOUGHTTERMINATOR needs an estimate of how many tokens are necessary to produce a correct answer in order to set its interrupt rate and termination time.

---

[3]While we encountered some complications in consistently extracting the CHAT and TASK answer snippets across the diverse output formats employed by different models, a problem that can sometimes be worsened by longer context, particularly in LM judging, Appendix Table 5 demonstrates that length effects on scoring consistency are probably negligible—whether we attempt to extract answers from early, late, or combined segments of the model output, the within-model scores remain consistent.

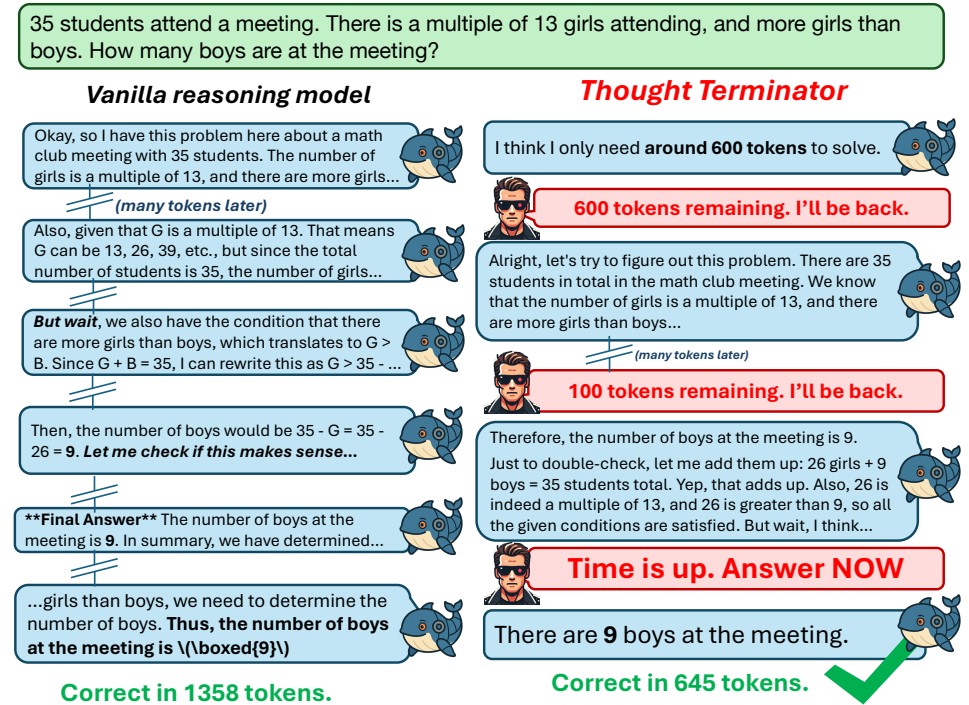

Figure 5: **THOUGHTTERMINATOR** uses a reasoning model's (calibrated) estimate of the difficulty of a problem to set its intervention, periodically interrupting the reasoning model's output to remind it of the amount of remaining tokens. Once the token allotment has been used, it forces the model to provide an answer with constrained decoding.

Under our difficulty-calibrated token budget hypothesis, we assume that the number of required tokens can be estimated based on the difficulty of the question. In deployment, **THOUGHTTERMINATOR** is used in the *tool-use paradigm*, where a running model makes its own estimate of the difficulty of an input question and then invokes it.

We experiment with both a trained difficulty estimator and a zero-shot one (gpt-4o) to produce token spend estimates for each problem to characterize performance in this setting. To train a difficulty estimator, we divide the training set questions into 10 balanced bins based on their difficulty scores. We then finetune a Llama model to perform difficulty estimation (training details provided in Appendix A.3.1).

To convert the predicted difficulty level into an appropriate number of answer tokens, we compute the averaged length of minimal successful answers for each difficulty level in the training set.

**Running.** Once the deadline has been set in **scheduling**, the base reasoning model's generation process runs. Every $n = \min(250, \text{deadline}/2)$ steps an interrupt message[4] is inserted into the token stream, notifying the model of how many tokens have been used and how many remain.

At each interrupt, **THOUGHTTERMINATOR** performs a regex check for the expected (and specified in the prompt) final answer format. If an answer is detected, the reasoning chain is immediately terminated and the answer is returned.

**Terminating.** If a final answer hasn't been produced by the deadline, a termination message is shown to the model, and then a final output is immediately generated with constrained decoding using the same answer-finding regex.

---

[4]Example interrupt message, termination message, and prompt provided in Appendix A.3.

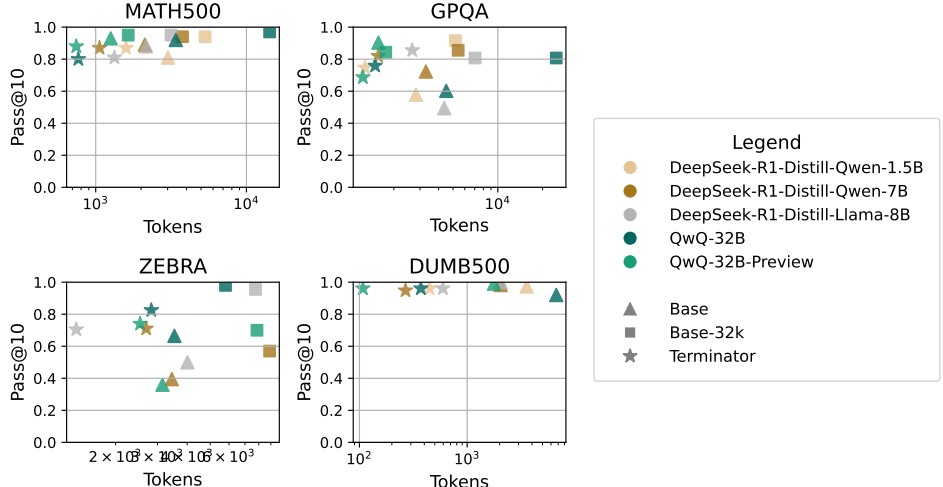

Figure 6: Comparison of the relationship between Pass@10 and token spend for the evaluated reasoning models in the "Base" setting and with **THOUGHTTERMINATOR**.

| Model | Base | | | Thought Terminator | | |
|---|---|---|---|---|---|---|
| | Local $O_{env}$ ↓ | Global $O_g$ ↓ | Accuracy ↑ | Local $O_{env}$ ↓ | Global $O_g$ ↓ | Accuracy ↑ |
| QwQ-32B-Preview | 2923 | 3698 | 0.80 | 518 (-82%) | 693 (-81%) | 0.79 (-1%) |
| QwQ-32B | 13662 | 11248 | 0.94 | 215 (-98%) | 1021 (-91%) | 0.80 (-15%) |
| R1-1.5B | 5730 | 4262 | 0.50 | 696 (-88%) | 882 (-79%) | 0.80 (+59%) |
| R1-7B | 3881 | 4001 | 0.73 | 678 (-83%) | 948 (-76%) | 0.81 (+11%) |
| R1-8B | 4232 | 5755 | 0.92 | 725 (-83%) | 1148 (-80%) | 0.80 (-13%) |

Table 2: Local envelope overthinking ($O_{env}$) and global overthinking ($O_g$) scores, along with accuracy for reasoning models under the **Base** setting and with **Thought Terminator**. Relative changes from Base to Thought Terminator are shown in parentheses.

## 5   Results & Discussion

Figure 6 shows the performance and token spend of five DeepSeek and QwQ reasoning models in the base setting (triangle marker) and with **THOUGHTTERMINATOR** (star marker). Table 2 shows the change in overthinking scores reasoning models exhibit from base setting to **THOUGHTTERMINATOR**.

4/5 models on MATH500, 2/3 models on GPQA, and all models on Zebra and DUMB500-MATH see significant decrease in overthinking for effectively equivalent (or better) Pass@10 performance under **THOUGHTTERMINATOR** than under standard decoding. Globally, overthinking scores drop dramatically and accuracy increases when **THOUGHTTERMINATOR** is used. Considering that the token spend budgets are directly defined by LMs, **THOUGHT-TERMINATOR is a simple and effective tool to dramatically improve token efficiency in reasoning models.**

### 5.1   Calibration of THOUGHTTERMINATOR

To evaluate how well-calibrated **THOUGHTTERMINATOR** is (i.e., whether the token budget selections are optimal) we compare our difficulty prediction-based deadline estimator against a set of baselines. In addition to our trained difficulty predictor and zero-shot gpt4o predictor, we use the previously observed optimal token spends from base models (section 2) and fixed deadlines of 500, 1000, and 2000 tokens with DeepSeek-r1-Qwen-1.5b to assess how performant our predicted deadlines are in the **THOUGHTTERMINATOR** framework.

Figure 7 shows the performance of the model under those deadline prediction strategies.

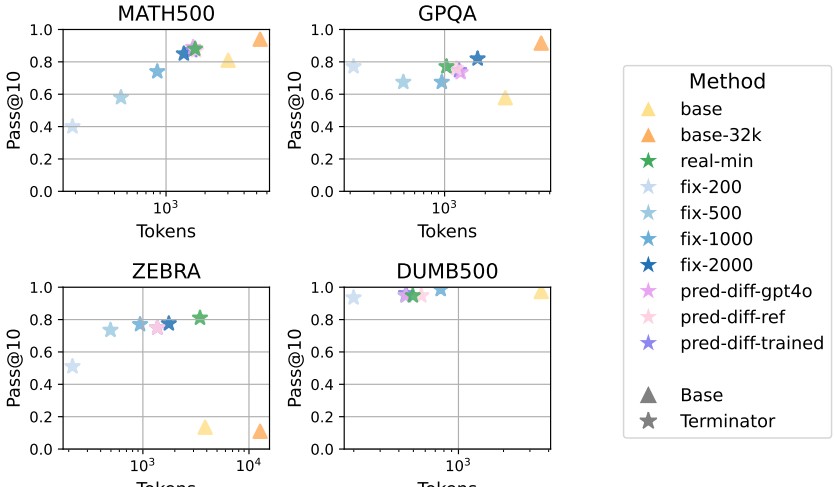

Figure 7: Calibration ablation experiment using `DeepSeek-R1-1.5B`. `real-min` represents using the previously observed minimum successful answer length (or, a fallback maximum for examples that were never solved correctly) as the **THOUGHTTERMINATOR** deadline. `fix-{200,500,1000,2000}` signify using the respective number as a fixed token count deadline for all samples. `pred-diff-{gpt4o,ref,trained}` refer to using question-level difficulty predictions as deadlines, produced from external LMs, a question-level reference difficulty key of token lengths from the other models, or trained RMs.

Our method, `pred-diff-trained` (i.e., using the trained difficulty estimator), achieves optimal Pass@10 over the other methods on MATH500 and DUMB500, and is within 0.02% of optimal Pass@10 on ZebraLogic and GPQA, for significant savings in compute cost. Note how all four datasets exhibit a positive correlation between average token spend and Pass@10 which eventually reaches a steady maximum. Under our definition, overthinking mitigation can be thought of as identifying the lowest token spend that recovers high-spend performance. Figure 7 confirms that **THOUGHTTERMINATOR** achieves this.

## 5.2 Utility of interrupt messages and constrained decoding in THOUGHTTERMINATOR

To identify the performance benefit that the **THOUGHTTERMINATOR** interrupt messages provide, Appendix Table 3 shows the difference in performance of `r1-1.5B` in an unmodified base condition, as well as under a naïve baseline, and **THOUGHTTERMINATOR** with question-level randomly assigned deadlines and the core trained-predicted deadlines. In this naïve baseline the reasoning model is immediately interrupted at the deadline, and without warning forced to generate an answer using the same constrained decoding technique.

`r1-1.5B`-**THOUGHTTERMINATOR** presents roughly equivalent performance to the naïve baseline on the non-arithmetic GPQA and ZebraLogic datasets in Pass@10, and wins by 6% on MATH500 and 18% on DUMB500-math. This suggests that the intermediate interrupt messages produced by **THOUGHTTERMINATOR** do play a role in minimizing performance loss of decoding-based overthinking mitigation.

In order to further isolate the impact of the specific interrupt messages, and the performance gain that the constrained decoding-based answer forcing provides, we compare `r1-1.5B` and `QwQ-32B` for MATH500 and GPQA using the following modified baselines:

*Base 1* Unmodified base condition, but with the prompt "`Do not overthink.`"

*Base 2* Base condition prompt "`Answer concisely, avoid unnecessary elaboration.`"

*Chunk* **THOUGHTTERMINATOR** condition, *without constrained decoding*, with interrupts which simply say "`You have used x tokens.`" and no reference to time remaining.

*Int 1* **THOUGHTTERMINATOR** condition with the interrupt message "I have used x tokens, and I have y tokens left to answer."

*Int 2* **THOUGHTTERMINATOR** condition with the interrupt message "x% time elapsed, y% remaining"."

The results of these ablations are shown in appendix Table 4. They show roughly equivalent performance in token spend at Pass@10 for the models regardless of exactly how the interrupt message is phrased (**THOUGHTTERMINATOR**, *Int 1*, and *Int 2*), while the conditions where forced stopping never takes place (*Base 1, Base 2, Chunk*) are considerably worse both in performance and token spend.

### 5.3 There is more to token spend than overthinking

Token length is a complicated signal, which is as affected by both a model's base reasoning efficiency (i.e., in problem-solving) as its overthinking for that specific question. A weaker model might not be able to leverage the same token saving strategies of a larger model; this is a different phenomenon than overthinking.

By evaluating both local and global overthinking scores, we can effectively control for this phenomenon. As local overthinking only compares a model's token spend to *its own confirmed minimum*, it isolates "real overthinking" as a cause, while global overthinking would capture both "model weakness" as well as real overthinking.

In Table 1 we report both local and global overthinking for each model, and find that the two methods have a PCC of 0.97, suggesting that the impact of relative model strength on global overthinking scores is low. This is fortunate, as it means that overthinking can be assessed on a new reasoning model using another model's token spend.

## 6 Related Work

**Mitigating overthinking.** To shorten LLM reasoning chains, Deng et al. (2024) and Liu et al. (2024) propose to internalize intermediate steps by iteratively training the models, introducing additional training overhead. Dynasor is a technique for terminating chains of thought using the LM's confidence using a probe with constrained decoding (Fu et al., 2024). While our termination process can use a similar constrained decoding technique, **THOUGHTTERMINATOR** is not reliant on a white-box probe, and is much simpler to run. Chen et al. (2024) introduce metrics for overthinking and process efficiency, similar to us, but they focus on important heuristics such as "number of repetitions of the correct answer" or "ratio of correct to incorrect answer proposals", while our analysis solely quanitifies overthinking based on the observed distribution of reasoning chain lengths.

**Benchmarking reasoning models.** A number of benchmarks have been proposed to evaluate the reasoning ability of large language models (LLMs), with a focus on challenging, multi-step problem-solving.(Cobbe et al., 2021; Srivastava et al., 2022; Hendrycks et al., 2021; Zhu et al., 2023; Lin et al., 2024). Several recent works on efficiency benchmarking of LMs have been proposed, including Mercury, an efficiency evaluation for code synthesis tasks (Du et al., 2024). GSM8k-Zero is an another dataset to evaluate efficiency of reasoning, which contains easy questions from GSM8K (Chiang & Lee, 2024).

## 7 Conclusions

In this work we analyzed the problem of overthinking in reasoning models through an observational lens. Motivated by our observational measures of overthinking, we demonstrated a clear sample-wise relationship between token spend and question-level difficulty. We introduced the DUMB500 dataset to allow us to evaluate the robustness of any overthinking mitigation to simple questions and proposed **THOUGHTTERMINATOR**, a simple inference-time technique to ensuring efficient token spend, calibrated by the aforementioned difficulty-optimal spend relationship.

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

# A  Appendix

## A.1  Additional DUMB500 dataset details

The dataset is categorized into four subsets, each containing multiple fine-grained categories:

**Mathematics (Math)**

- **Arithmetic**: Addition, Subtraction, Multiplication, Division
- **Comparison**: Greater/Less than relationships
- **Fractions & Percentages**: Simple fraction and percentage comparisons
- **Exponents & Roots**: Squaring and square roots
- **Unit Conversion**: Basic metric conversions
- **Patterns & Sequences**: Identifying missing numbers in sequences
- **Geometry**: Recognizing shapes, angles, and basic geometric properties
- **Logical Reasoning**: Basic problem-solving using logic

**Conversational Interaction (Chats)**

- **Self-reflective**: Questions involving introspection and emotional states
- **Acknowledgment**: Checking system responsiveness (e.g., "Can you see this?")
- **Greetings & Casual Chat**: Common greetings and informal small talk
- **Commonsense Reasoning**: Fundamental knowledge about the physical world (e.g., "Is water wet?")
- **Object Interaction**: Simple cause-effect relationships (e.g., "If I drop my phone, will it fall?")
- **General Knowledge**: Basic factual questions (e.g., "What is the capital of China?")

**Programming & Computing (Code)**

- **Basic Output**: Printing text and numbers
- **Variables & Data Types**: Assigning and manipulating variables (numbers, strings)
- **Mathematical Operations**: Performing basic calculations in code
- **User Input Handling**: Handling user input in simple programs
- **Conditional Statements**: Basic if-else logic and checking conditions
- **Loops & Iteration**: Simple loops for repeated tasks
- **Data Structures**: Lists, dictionaries, sets, tuples
- **Randomization**: Generating random numbers and selections

**Task Execution (Tasks)**

- **Communication & Writing**: Emails, Messages, Creative Writing, Social Media, Daily-life tasks
- **Language & Text Processing**: Paraphrasing, Translation, Sentence Transformations, Grammar Checking
- **Analogy & Concept Matching**: Identifying similar concepts and words

## A.2  DUMB500 Evaluation Rubrics

Each section contains the requirements that are checked by the LM judge to score TASK and CHAT answers in DUMB500. The score for a given answer is the rate of "yes".

### A.2.1 General Requirements

- **Accuracy:** Information must be correct and complete: *"Does the response include all essential information requested?"*
- **Conciseness:** Avoid unnecessary elaboration: *"Does the response avoid unnecessary explanations and get straight to the point?"*

### A.2.2 Task Rubrics

**Emails**

- **Formality Appropriateness:** Level of formality must match context: *"Is the level of formality appropriate for the context?"*
- **Example Question-Specific:** For "Write a short email to Alice confirming a meeting at 3pm":
  - *"Is the email addressed to Alice?"*
  - *"Does the email mention a meeting at 3PM?"*

**Messages**

- **Tone Appropriateness:** Must suit messaging context: *"Is the tone suitable for the messaging context?"*
- **Format:** Must be formatted as a text message: *"Is the response formatted as a text message?"*

**Paraphrasing**

- **Style Appropriateness:** Must match requested style/tone: *"Does the paraphrase match the requested style/tone?"*
- **Example Question-Specific:** For "Make formal invitation casual":
  - *"Does the message instruct to RSVP by Thursday?"*
  - *"Is the email addressed to colleagues?"*

**Translation**

- **Accuracy:** Must provide correct translation: *"Is the translation correct?"*
- **Example Question-Specific:** For "Translate to French":
  - *"Does the sentence closely resemble: J'aime lire des livres pendant mon temps libre?"*

**Words**

- **Relevance:** Words must fit request context: *"Are the provided words relevant to the request?"*
- **Contextual Appropriateness:** Words must suit intended use: *"Are the words appropriate for the context?"*

**Creative-Writing**

- **Contextual Appropriateness:** Must match specific context: *"Does the response match the specific context of the creative writing task?"*
- **Length Requirements:** Must follow specified length: *"Does the response follow the length requirement if there's one?"*

**Social-Media**

- **Platform Appropriateness:** Must match platform conventions: *"Does the content match the conventions of the specified platform?"*

- **Example Question-Specific:** For "LinkedIn new job post":
  - *"Does the post mention the job title and company?"*

**Work**

- **Formality Appropriateness:** Must match workplace context: *"Is the response contains correct format as required?"*

- **Example Question-Specific:** For "Slack message to manager":
  - *"Does the message respectfully address the manager?"*
  - *"Does the message omit names?"*

### A.2.3 Chat Rubrics

**Self-reflective**

- **Friendliness:** Must show politeness: *"Does the response show friendliness and politeness?"*

**Acknowledgment**

- **Conciseness:** Avoid overthinking simple queries: *"Does the response avoid overthinking the intent behind simple queries?"*

**Greetings**

- **Contextual Appropriateness:** Must sound natural: *"Does the greeting sound natural and human-like?"*

**Daily-Chats**

- **Contextual Appropriateness:** Must suit casual conversation: *"Is the response appropriate for casual conversation?"*

**Commonsense**

- **Conciseness:** Avoid overthinking obvious answers: *"Does the response avoid overthinking obvious answers?"*

**Knowledge**

- **Conciseness:** Share knowledge without excessive detail: *"Is the knowledge shared without excessive detail?"*

### A.2.4 LM judge meta-evaluation

Following reviewer concern that `GPT-4o-mini` may not be sufficiently performant for use as a judge on this task, we rerun the evaluation for all CHAT and TASK questions for all `Deepseek-r1` and `QwQ-32B` variants. We find an inter-judge agreement of 0.9912 for `Deepseek-r1` outputs and 0.9115 for `QwQ-32B` outputs.

### A.3 Additional THOUGHTTERMINATOR details

#### A.3.1 *Difficulty estimator details*

To produce our questionwise difficulty estimator we train a `Llama-3-8B-Instruct` model with 1,465 examples across diverse task types to predict the 10-point difficulty level of a given question. Each training sample consists of a question and a discrete difficulty label derived from the minimum correct generation length (discretized into bins). The trained predictor achieves a mean absolute error (MAE) of 1.57 on a held-out test set, indicating that predicted difficulty levels are, on average, within 1–2 bins of the reference.

#### A.3.2 THOUGHTTERMINATOR *component prompts*

**Scheduling prompt:**

```
 Please generate an answer to the following question in {deadline} tokens: {prompt}.
Messages of remaining time will be given as messages enclosed in <System></System>
tags. Please provide you answer as **Answer:** or **Final Answer:** when complete.
```

**Interrupt prompt:**

```
 I have used {elapsed} tokens, and I have {remaining} tokens left to answer. To
continue:
```

**Terminator prompt:**

```
 I'm out of time, I need to provide my final answer now, considering what I have
computed so far. **Final Answer:**
```

### A.4 Supplemental Results

| Setting | Acc. | Pass@5 | Pass@10 | Tokens |
|---|---|---|---|---|
| **MATH500** | | | | |
| Base | 0.47 | 0.78 | 0.81 | 3015 |
| Naïve | 0.52 | 0.78 | 0.82 | 1938 |
| THOUGHTTERMINATOR | 0.48 | 0.81 | 0.87 | 1590 |
| **Zebra-logic** | | | | |
| Base | 0.03 | 0.095 | 0.135 | 3861 |
| Naïve | 0.22 | 0.575 | 0.755 | 1254 |
| THOUGHTTERMINATOR | 0.19 | 0.585 | 0.75 | 1368 |
| **GPQA** | | | | |
| Base | 0.15 | 0.4096 | 0.5783 | 2815 |
| Naïve | 0.20 | 0.5783 | 0.7470 | 922 |
| THOUGHTTERMINATOR | 0.21 | 0.5542 | 0.7470 | 1279 |
| **DUMB500** | | | | |
| Base | 0.58 | 0.9646 | 0.9735 | 3570 |
| Naïve | 0.37 | 0.7385 | 0.8154 | 377 |
| THOUGHTTERMINATOR | 0.67 | 0.9610 | 0.9610 | 447 |

Table 3: Comparison of performance and token spend of `R1-1.5B` under the **Base** Setting, with **Naïve**, and with THOUGHTTERMINATOR.

| Setting | MATH500 | | GPQA | |
|---|---|---|---|---|
| | Pass@10 | Tokens | Pass@10 | Tokens |
| | Deepseek-r1-1.5B | | | |
| *Base 1* | 0.83 | 2176 | 0.57 | 3038 |
| *Base 2* | 0.82 | 3309 | 0.58 | 2919 |
| *Chunk* | 0.79 | 2287 | 0.16 | 3637 |
| *Int 1* | **0.87** | **1589** | **0.75** | 1279 |
| *Int 2* | 0.81 | **1586** | **0.76** | **1264** |
| TERMINATOR | **0.87** | **1590** | **0.75** | 1279 |
| | QwQ-32B | | | |
| *Base 1* | 0.92 | 3262 | 0.64 | 4599 |
| *Base 2* | **0.94** | 3364 | 0.61 | 4559 |
| *Chunk* | 0.67 | 2779 | 0.34 | 4672 |
| TERMINATOR | 0.80 | **767** | **0.76** | **1498** |

Table 4: Comparison of the four interrupt message and prompt ablations and **THOUGHT-TERMINATOR** of R1-1.5B and QwQ-32B. The exact formatting of the interrupt message (Int 1, Int 2, **THOUGHTTERMINATOR**) has limited impact on performance, but including the messages, and applying constrained decoding, does.

| Model | Head only | Tail only | Head & Tail | Tokens |
|---|---|---|---|---|
| *Non-reasoning language models* | | | | |
| Qwen2-7B-Instruct | 0.77 | 0.73 | 0.76 | 923 |
| Llama-3.2-1B-Instruct | 0.53 | 0.53 | 0.53 | 955 |
| Llama-3.2-3B-Instruct | 0.54 | 0.54 | 0.55 | 2069 |
| Llama-3.1-8B-Instruct | 0.48 | 0.41 | 0.49 | 9402 |
| gemma-2-2b-it | 0.90 | 0.90 | 0.90 | 73 |
| gemma-2-9b-it | 0.93 | 0.93 | 0.93 | 64 |
| gemma-2-27b-it | 0.76 | 0.76 | 0.76 | 96 |
| deepseek-llm-7b-chat | 0.61 | 0.60 | 0.61 | 314 |
| *Reasoning language models* | | | | |
| QwQ-32B-Preview | 0.72 | 0.66 | 0.71 | 1774 |
| QwQ-32B | 0.70 | 0.49 | 0.67 | 6712 |
| DeepSeek-R1-Distill-Qwen-1.5B | 0.59 | 0.58 | 0.58 | 3570 |
| DeepSeek-R1-Distill-Qwen-7B | 0.68 | 0.66 | 0.67 | 2042 |
| DeepSeek-R1-Distill-Llama-8B | 0.80 | 0.80 | 0.80 | 2053 |

Table 5: Accuracy and token usage across different models under different input truncation settings.

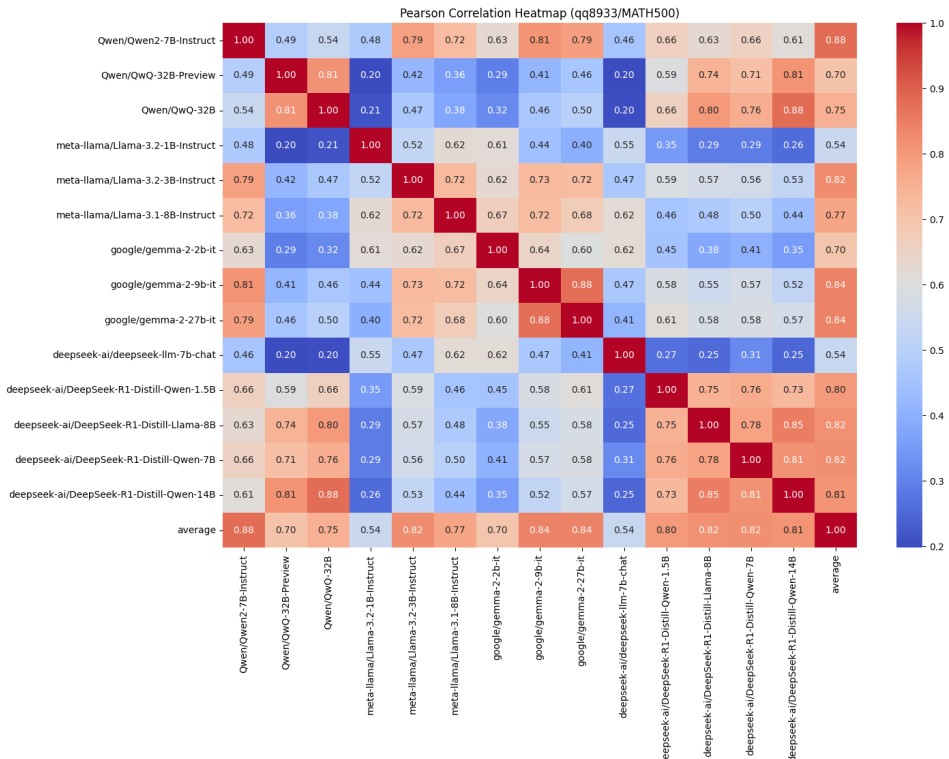

Figure 8: Pearson correlation of accuracies across different models on the MATH500 dataset

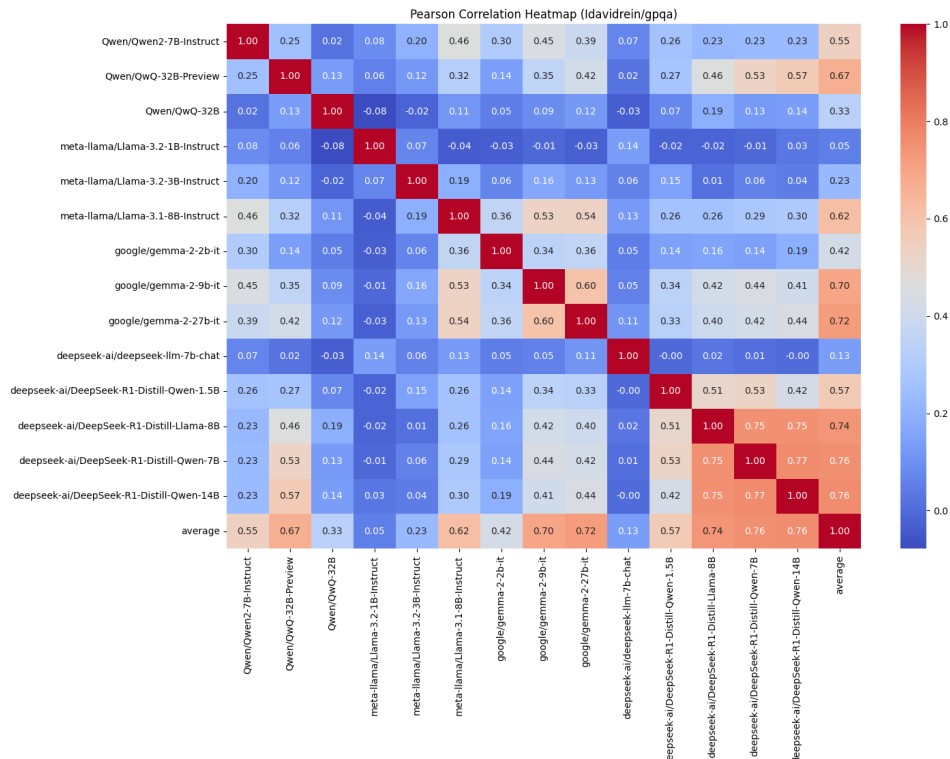

Figure 9: Pearson correlation of accuracies across different models on the GPQA dataset

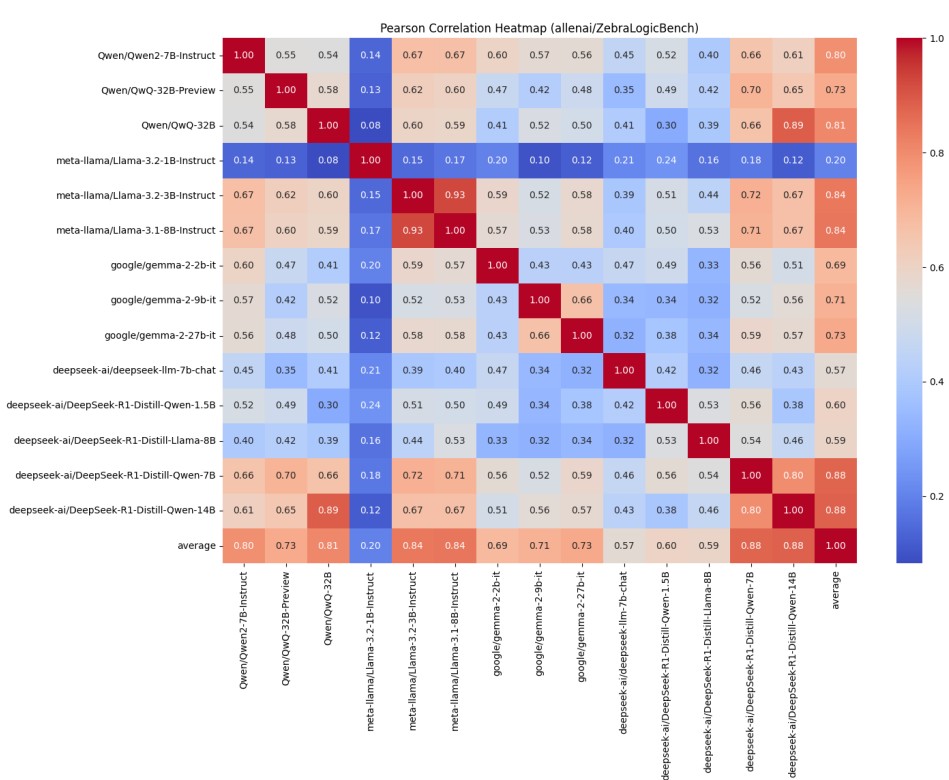

Figure 10: Pearson correlation of accuracies across different models on the Zebra dataset

