# OpenReview forum: "ThoughtTerminator: Benchmarking, Calibrating, and Mitigating Overthinking in Reasoning Models"
_colmweb.org/COLM/2025/Conference — COLM 2025_

### Official Review · Reviewer_U1jN · 2025-04-27

**Rating:** 6
**Confidence:** 5
**Ethics Flag:** 1

**Summary:**

This work has three main contributions:
+ It proposes measuring problem-level difficulty by the model’s output token length. Based on this, the paper introduces a method to quantify the severity of a model’s overthinking, using both the model output token length and the problem difficulty.
+ It introduces DUMB500, a dataset of extremely easy problems. A model that overthinks is expected to answer problems in this dataset without generating too many output tokens.
+ It proposes a decoding method, THOUGHTTERMINATOR, to improve token efficiency.

**Questions To Authors:**

+ Since different models may use different tokenizers, how do you ensure fair comparisons in Table 1? Also, in Formula 3, token lengths from different models may not be directly comparable or simply averaged. Although this might be a minor issue, clarification would be helpful.
+ In Figure 4, why do all the models perform poorly on Code, Task, and Chat? As you claimed, DUMB500 should consist of very easy problems. I had previously assumed that DUMB500 is designed so that models can easily solve the problems and the main challenge (for a model that tends to be overthinking) is minimizing token usage. However, the observed poor performance suggests that my assumption might be incorrect. Could you clarify this?
+ What do all the baselines shown in the legend of Figure 7 represent? I could not find their definitions in the main text. Specifically, what do `real-min` and `pred-diff-ref` mean?

**Reasons To Accept:**

+ The research problem studied in this work is very important. Recently, it has been increasingly recognized that overthinking in strong reasoning LLMs is a serious issue.
+ The DUMB500 dataset is interesting and seems to be a potentially useful testbed for future work on LLM overthinking.
+ The proposed decoding method, THOUGHTTERMINATOR, appears to be effective. It also does not require any training, which could make it easily pluggable into different models.

**Reasons To Reject:**

+ I am not fully convinced that simply measuring token length is a reliable way to quantify the severity of overthinking. Token length itself seems to be a complicated signal, coupling both the model’s reasoning capability (i.e., problem-solving ability) and overthinking. For example, a weaker model might be unable to come up with a smart solution for a difficult math problem and thus has to brute-force a solution, resulting in a much longer token output compared to a stronger model that leverages a clever trick to simplify computations. In this case, it would not make sense to attribute the longer token output to overthinking, as it simply reflects the weaker model’s limited problem-solving capability.
+ The paper does not compare THOUGHTTERMINATOR with a simpler baseline:
   - For instance, instead of estimating the required number of tokens externally, we could instruct the reasoning LLM itself (via system prompts) to try its best to save tokens.
   - Additionally, when generating every CHUNK of tokens, we could append a reminder to the context such as "You have consumed {CHUNK × TIMES} tokens" to make the model aware of its token usage.
   - In this way, we leverage the reasoning LLM’s own capability to estimate the token length needed, which might be more effective than relying on an external model (e.g., a LLaMA-based predictor or GPT-4o), which may not closely match the internal behavior of the target model.
+ What are the main challenges in collecting DUMB5000? My assumption is that one could simply compile older LLM benchmarks that have become easy for today’s models. Please correct me if this assumption is incorrect.
+ I feel that the writing could be improved:
   - Overall, the paper’s structure could be more coherent. The three main contributions currently feel somewhat disconnected—would it be possible to connect them into a more unified narrative?
   - Additionally, some figures (which I will discuss further in the Questions section) seem unfinished. For example, some legends are unclear or hard to interpret.

---

> ### Author Response · Authors · 2025-06-03
>
> Thank you for your insightful and detailed review.
>
> ## Weaknesses
>
> 1. Token length isn’t explained by overthinking alone, relative model weakness also matters
>
> Great point. This is why we analyze both local and global overthinking. *Local overthinking* only compares reasoning chains *within model*; comparing the distribution of reasoning chain lengths for each question to the observed minimum successful length for that model. For a weaker model which requires brute forcing *longer reasoning chains than necessary still contain overthinking*. In other words local overthinking will capture “real overthinking” in this case while global overthinking would capture both “model weakness” as well as real overthinking.
>
> In Table 1 we report both local and global overthinking for each model. **The two methods have a PCC of 0.97**, suggesting that the impact of relative model strength on global overthinking scores is low.
>
> We will include your point, as well as this response, in the discussion section of our CR. Overthinking in RMs has yet to gain a consensus definition, so it is important that we precisely convey what implications and potential problems lie in our operationalization, and this point about the correlation between the two scores strengthens our overall argument.
>
> ---
>
> 2. Simpler baseline suggestions
>
> These are interesting baseline proposals. Re: your first proposal, we add two baselines:
>
> - Baseline 1: Adds the instruction “Answer concisely and avoid unnecessary elaboration.”
> - Baseline 2: Adds the instruction “Do not overthink.”
>
> We evaluated these along with the base setting and our method (ThoughtTerminator) on two models (R1-1.5B and QwQ-32B) across two datasets (MATH500 and GPQA). The results are shown below:
>
> | Model | Dataset | Base (P@10 / tokens) | Baseline 1 | Baseline 2 | Terminator |
> |-----|---|----|------|-----|----|
> | R1-1.5B  | MATH500 | 0.81 / 3015.2  | 0.83 / 2176.0   | 0.82 / 3309.7  | 0.87 / 1589.9 |
> |  | GPQA | 0.58 / 2815.5 | 0.57 / 3038.7  | 0.58 / 2919.4   | 0.75 / 1279.0  |
> | QwQ-32B  | MATH500 | 0.92 / 3399.4  | 0.92 / 3262.3 | 0.94 / 3363.8 | 0.80 / 767.1 |
> | | GPQA | 0.60 / 4502.0 | 0.64 / 4599.1 | 0.61 / 4558.9 | 0.76 / 1497.8 |
>
> These baselines, while sometimes outperforming the base condition, are beaten clearly by Terminator.
>
> We will leave the other proposed baselines for the CR.
>
> ---
>
> 3. Do older baselines contain questions as easy as DUMB500?
>
> We found that for the math questions, no extant dataset was as simple as we wanted. We wanted to expand out a set of questions of similar difficulty to the canonical “what does 2+2 equal?” example, which were too easy to be worth including in prior datasets, even for the most rudimentary ones 5 years ago. By the time researchers were bothering to evaluate LMs on math, they were already good enough to get such questions right.
>
> ---
>
> 4. Narrative structure feels incoherent, figures seem unfinished
>
> We recognize that our paper has an unconventional structure. We felt that the traditional “problem statement->methods->results” format was inappropriate for this work, as the initial overthinking quantification results inform the need for the DUMB500 dataset, and then overthinking/accuracy results for the four datasets inform the design of Terminator. We believe this is the ideal structure, but will revise the introduction to more clearly broadcast the logic and layout to readers to reduce confusion.
> Re: figures, see answers to questions.
>
> ## Questions
> 1. Different tokenizers, different overthinking scores?
>
> This is a good question. We believe that the correlation between global and local overthinking discussed above suggests that this is a minor issue. However, in the CR we will provide an analysis where we compare the token lengths of a set of reference strings between each model, and recompute overthinking scores after normalizing based on these as an ablation.
>
> ---
>
> 2. Why are models bad at DUMB500 chats and tasks?
>
> Task and chat examples are actually graded using LM judges and numeric rubrics rather than accuracy, so average scores <<1 do not necessarily represent severely bad performance as they would on an accuracy scale. We have revised the axis labels to reflect this. Furthermore, we want to highlight that the math subset of DUMB500 is most central to our story, the other three partitions are mainly introduced to (a) enable future work and (b) show overthinking on diverse domains.
>
> ---
>
> 3. What do `real-min` and `pred-diff-ref` mean?
>
> Sorry for this oversight! `real-min` is an ablation where we simply fix the minimum observed token spend for each example to be the deadline, isolating the effect of applying Terminator by removing the influence of poor difficulty predictions for a given sample.  `pred-diff-ref` in turn is an ablation where the ground truth bucketed *difficulty level prediction* for each question is fixed, but the difficulty->token spend predictor is still used. We have clarified this explanation in the manuscript.

---

> > ### Comment · Reviewer_U1jN · 2025-06-03
> >
> > Thanks!
> >
> > In the baseline I proposed, there is also a component: `when generating every CHUNK of tokens, we could append a reminder to the context such as "You have consumed {CHUNK × TIMES} tokens" to make the model aware of its token usage.`

---

> > ### Author Response · Authors · 2025-06-09
> >
> > Here are results for the full baseline you suggested:
> >
> > | Model | Method | MATH500 (P@10 / tokens) | GPQA |
> > |-----|---|----|------|
> > | R1-1.5B  | {CHUNK x TIMES} | 0.79 / 2287.3 | 0.16 / 3637.3 |
> > |  | Terminator | **0.87** / *1589.9* | *0.75* / **1279.0** |
> > | QwQ-32B  | {CHUNK x TIMES} | 0.67 / 2779.2 | 0.34 / 4672.7 |
> > | | Terminator | *0.8*   / **767.1** | **0.76** / *1497.8* |
> >
> > This baseline is equivalent to Terminator, but with your proposed alternate intermediate messages and no forced answering with constrained decoding at a deadline.
> >
> > This result demonstrates that the reasoning models are not good at explicitly controlling their own token spend, which seems to align with observations from prior work such as s1.
> > During this run, we observed an interesting issue with GPQA in particular: both QwQ-32B and r1-1.5B would fail to generate a final answer. We hypothesize that over the non-math questions in GPQA the intermediate messages push the outputs to far OOD for natural EOS tokens and answers to be generated. By forcing an answer with constrained decoding at the end ThoughtTerminator mitigates this problem.
> >
> > Taking results of all ablations together, **the interrupt messages encourage efficiency in intermediate reasoning**, and **forced stopping with constrained decoding controls token spend**. By combining both of these elements, ThoughtTerminator performs the best for both minimizing token spend, and maximizing accuracy.
> >
> > We hope that in light of this additional ablation and our initial rebuttal you are open to revising your review.

---

> > > ### Comment · Reviewer_U1jN · 2025-06-09
> > >
> > > Sorry if I missed this - what would be the result if you only did forced stopping?
> > >
> > > One more question: how did you train the difficulty estimator? I was wondering how you can make sure it has good generalizability (sorry if i missed this in your paper)

---

> > ### Author Response · Authors · 2025-06-09
> >
> > Our mistake, the table explaining this baseline wasn't present in the submission version! We did perform this ablation using R1-1.5B on all four datasets.
> >
> > | Dataset | Setting |   Pass@10 | Tokens |
> > |---|---|---|----|
> > | MATH500 | Base |   0.81 |  3015 |
> > | | Early stopping | 0.82 | 1938 |
> > | | Terminator |  **0.87** | **1590** |
> > | Zebra-logic | Base | 0.135 | 3861 |
> > | | Early stopping | **0.755** | **1254** |
> > | | Terminator |  **0.75** | 1368 |
> > | GPQA | Base |  0.5783 | 2815 |
> > | | Early stopping | **0.7470** | 1922 |
> > | | Terminator | **0.7470** | **1279** |
> > | DUMB500 (Math) | Base | 0.9735 | 3570 |
> > | | Early stopping | 0.8154 |  377 |
> > | | Terminator | **0.9610** | **44** |
> >
> > Across all datasets (for R1-1.5B) Terminator ties or beats simple early stopping for Pass@10, and Terminator beats Early Stopping for token efficiency on all datasets except ZebraLogic.
> >
> >
> > ---------
> >
> > Regarding the trained difficulty estimator, apologies for a lack of clarity.
> >
> > All of the `pred-diff-*` ablation conditions in Figure 7 (including `pred-diff-trained` which is used in Terminator) involve a two-step process:
> > 1. Predict the difficulty bucket (10-point scale, see Figure 3) the question falls in
> > 2. From the difficulty bucket, predict the required token spend
> >
> > We use this two-step abstracted method for better generalization over a direct token prediction technique which we found too brittle for final use.
> >
> > To train the difficulty estimator and learn a mapping from bucket to token spend, we hold out 100 train samples from each dataset. For the mapping (2), we simply store the average token spend over correct runs for each model for each question in a given bucket.
> >
> > For difficulty predictors (1), we test three: `pred-diff-trained` uses a trained LLaMA model, `pred-diff-gpt4o` uses a few-shot prompt, and `pred-diff-ref` uses the known difficulty scores for all questions to control for the impact of the difficulty predictor and assess the impact of the difficulty->token budget predictor.
> >
> > Our trained difficulty estimator is a LoRA on LLaMA-3-8B-Instruct which trains for 3 epochs over the combined set of 1,465 questions sampled from the four datasets, with a balanced difficulty distribution to the remaining test questions.
> >
> > The ablation in Figure 7 shows how these three difficulty predictors cluster quite close together in the performance/token spend space. On MATH500 they achieve the same overall optimal performance, on GPQA they cluster together (but are suboptimal compared to the naive baseline of always setting the deadline to 2000), and similar for ZebraLogic and DUMB500. Because we see similar performance between the few-shot and trained difficulty estimation methods, we expect this part of the pipeline to be generalizable.
> >
> > Assessing the generalizability of the difficulty->token budget predictor is harder using our current set of experiments, and may be an interesting direction for future work. Beyond how well Terminator token spend predictions between tasks, generalization in reasoning models is an interesting open problem. (Anecdotally) it appears that current open (particularly any based on Qwen) are overfit to math reasoning tasks. As long as training in-distribution (either SFT or with RL) is a requirement for getting a good RM for some domain, Terminator-style difficulty prediction could be an added element in this training mix.
> >
> > ---------
> >
> > Once again, we are very grateful for your engagement throughout the discussion period!

---

> > > ### Comment · Reviewer_U1jN · 2025-06-09
> > >
> > > OK, thanks for your response. I am happy to slightly increase my score (from 5 to 6). That said, I think the paper should be improved in terms of writing and clarity; if it is accepted, I hope all of the details can be made clear in the camera-ready version.
> > >
> > > Another thing is that I kind of doubt if it makes sense to use a separate base model for the difficulty predictor - different models may have their own judgment of difficulty based on their own capabilities; maybe it is worth trying to use the same model as both the reasoning model and the difficulty predictor.

---

### Official Review · Reviewer_aNwS · 2025-05-12

**Rating:** 6
**Confidence:** 3
**Ethics Flag:** 1

**Summary:**

The paper focuses on the issue of overthinking of reasoning models. This work presents a new approximate measures of problem-level difficulty and demonstrates the relationship of difficulty and token spends. The finding reveals the calibration issue of reasoning models and its “overthinking” over easy problems. DUMB500 is a dataset introduced. Finally a decoding trick called ThoughTerminator was introduced.

**Questions To Authors:**

Can you explain the “A_hat ~ M !=a”? I don’t get the a_hat here.

I get a little confused by the percentage in Table 2. 0.58 -> 0.50, should it be 8%, if it’s 1.00 scaled.

**Reasons To Accept:**

Overthinking and the calibration of reasoning models is a timely research problem and this paper is well-fitted in this gap.

The paper proposes ToughtTerminator, a prompting + tool use approach to address the overthinking issue. It’s a simple enough and effective baseline and I believe it will be widely adopted.

The introduction of DUMB500 could be used to evaluate this issue.

There are many interesting findings for a range of frontier reasoning models. These findings about token spend, accuracy and optimal usage could benefit and inspire the community.

**Reasons To Reject:**

There are some baselines and settings not explored in this paper. For instance, adding a prompt like “think accordingly / think as you need / do not overthink” or adjust the sampling parameters. I think it would be essential to deploy some easy fix before claiming it’s a problem.

There are some key assumptions and I am not sure if they are rigorous. For example, the difficulty of a task is the accuracy of tested models. The overthinking score is the mean diff between the length of each reasoning chain and the global observed min spend for each question. These metrics depend on the choice of models. If there is one specific model never thinks, the score will drastically change. The definition of “overthinking score” should be more intrinsic rather than relative evaluation.

---

> ### Author Response · Authors · 2025-06-03
>
> 1. On more baselines for comparison
>
> We thank the reviewer for this suggestion.
>
> Following the feedback, we **added two prompt-based baselines**:
>
> - Baseline 1: Adds the instruction “Answer concisely and avoid unnecessary elaboration.”
> - Baseline 2: Adds the instruction “Do not overthink.”
>
> We evaluated these along with the base setting and our method (ThoughtTerminator) on two models (R1-1.5B and QwQ-32B) across two datasets (MATH500 and GPQA).
> The results are shown below:
> | Model | Dataset | Base (P@10 / tokens) | Baseline 1 | Baseline 2 | Terminator |
> |----------|---------|----------------------|------------------------|------------------------|------------------------|
> | R1-1.5B | MATH500 | 0.81 / 3015.2 | 0.83 / 2176.0 | 0.82 / 3309.7 | 0.87 / 1589.9 |
> | | GPQA | 0.58 / 2815.5 | 0.57 / 3038.7 | 0.58 / 2919.4 | 0.75 / 1279.0 |
> | QwQ-32B | MATH500 | 0.92 / 3399.4 | 0.92 / 3262.3 | 0.94 / 3363.8 | 0.80 / 767.1 |
> | | GPQA | 0.60 / 4502.0 | 0.64 / 4599.1 | 0.61 / 4558.9 | 0.76 / 1497.8 |
>
> These results show that:
>
>  - The Terminator consistently **achieves much lower token usage**, often by more than 50%, compared to all baselines;
>  - In most cases, it also **improves or maintains performance**, especially on GPQA;
>  - These baselines offer modest improvements in a few cases, but they do not reliably reduce overthinking or token inefficiency.
>
>  We believe these experiments confirm that **ThoughtTerminator is more effective and consistent than simple prompt-based heuristics**.
>
>  ---
>
>  2. On the assumptions behind difficulty and overthinking scores
>
>  2.1 Why we chose an empirical definition
>
>  Our difficulty and overthinking scores are **empirical**, i.e., they reflect how today’s models behave in practice.
>  Our choice aligns with prior work like Dynabench [1], which also embraces an empirical, model-relative view of task difficulty.
>
>  We use observed model behavior as the reference:
>
>  - Difficulty – average error rate across a diverse pool of 12 models (non-reasoning + reasoning). This captures how hard today’s models find the question, even if the theoretical difficulty is low.
>  - Overthinking score – excess tokens relative to the shortest successful trace seen anywhere. This tells us how far a given model is from the best practice we have observed in the current ecosystem.
>
>  This design intentionally reflects practical rather than intrinsic efficiency.
>
>  2.2 Robustness to the choice of model pool
>
>  > If there is one specific model never thinks, the score will drastically change.
>
>  Our paper introduces two overthinking metrics:
>
>  - Global Overthinking Score, which compares each model’s token usage to the minimum across all models (i.e., global baseline), and
>  - Local Envelope Score, which only uses a single model's own behavior and thus is independent of other models.
>
>  We agree that the **absolute value of the Global Overthinking Score may vary depending on the model pool**—for instance, if one model always uses extremely few tokens, it can lower the global baseline and affect others’ scores. However, we emphasize that this score is **meant to capture relative differences across models**, not serve as an intrinsic measure.
>
>  To test robustness, we progressively removed the models with the shortest average successful outputs (top 1, top 2, and top 3). In all cases, **the ranking of models by global overthinking score remained unchanged**. This indicates that while the baseline may shift slightly, the comparative insights drawn from the scores are robust.
>
>  This supports the view that our proposed overthinking scores produces stable rankings and comparisons, even if absolute values shift slightly with the model pool.
>
>  ---
>
>  3. On “A_hat ~ M ≠ a” Notation
>
>  > Can you explain the “A_hat ~ M !=a”? I don’t get the a_hat here.
>
>  Thank you for pointing this out. We agree the notation was unclear. In our intended formulation,
>
>  - $\hat{a}_M(q)$ denotes the model $M$’s predicted answer to question $q$,
>  - $a$ is the correct (gold) answer.
>
>  The expression “A_hat ~ M != a” simply means that the model's answer is incorrect. We will add an explanation of $a_hat$ in our manuscript.
>
>  ---
>
>  4. On Percentage Change in Table 2
>
>  To clarify: as we mentioned in the table caption, all percentage changes reported in Table 2 are relative improvements. For example, a drop from 0.58 to 0.50 is reported as (0.50 - 0.58)/0.58 = -13.8%.
>
>  ---
>
>  References: [1] Douwe Kiela et al. Dynabench: Rethinking Benchmarking in NLP. In Proceedings of the 2021 Conference of the North American Chapter of the Association for Computational Linguistics

---

> ### Author Response · Authors · 2025-06-09
>
> Hi Reviewer aNwS, we just wanted to check if you have any thoughts or concerns regarding our rebuttal. Thanks!

---

### Official Review · Reviewer_1isg · 2025-05-12

**Rating:** 6
**Confidence:** 4
**Ethics Flag:** 1

**Summary:**

First, thanks to the authors for their work. This paper attempts to first quantity, then mitigate the overthinking problem of current LLMs. The overthinking problem is defined as generating unnecessary tokens during inference which don’t improve the accuracy of a question. The authors take the following actions to tackle this problem:
- They introduce a new way to estimate the difficulty of a task, by sampling n answers for a given question and computing the rate of inaccurate answers.
- They quantify overthinking by using the difficulty of a question and the observed token spend per question.
--Global overthinking score for a given model - the difference between the mean spend tokens for a question by the given model and the absolute minimum observed spend across different models for the same question.
--Local envelope overthinking score for a given model - the mean difference between the maximum observed spend for a question and the minimum observed spend for the same question by the same model.
-Introduced a new dataset called DUMB500, consisting of simple questions in four domains: mathematics, conversational interaction, programming & computing, task execution, aimed to measure the overthinking of the models on dumb/easy questions.
-Introduced the THOUGHTTERMINATOR, a simple text-augmentation method aimed to reduce the general overthinking of models.
--Doesn’t require re-training of the model.
--Consists of three stages:
---Scheduling - estimates the number of tokens to solve a given question.
---Running - sends/inserts interrupt messages to notify the model of how many tokens have been remaining.
---Terminating - if a final answer hasn’t been produced when all the tokens have been spent, sends a termination message asking the model to produce the final answer.
-The results of THOUGHTTERMINATOR are presented on 4 datasets (including DUMB500), using 5 different models, showing improvements.

**Questions To Authors:**

- As a small suggestion, can you improve the readability of section 2.1, where the overthinking measures are introduced? It took me a few reads back and forth between the formulas and text to understand the metrics.
- In figure 7, for the ZEBRA dataset, the base seems to use the most tokens but produce the worst (by a huge margin) result. This seems like an outlier, and it  wasn't addressed in the text. Can you please address it?
- Will the DUMB500 dataset be available publicly?

**Reasons To Accept:**

- The novelty of estimating the difficulty of a task.
- The novelty of quantifying a model's overthinking.
- The introduction of the DUMB500 dataset.
- The introduction of THOUGHTTERMINATOR, and the showed improvements of using it.

**Reasons To Reject:**

- More ablations for the THOUGHTTERMINATOR are needed. The only provided (and as far as the public is aware, tried) prompt is the one presented in the paper. Since we have seen in the past evidence that the models can’t reliably count letters, count words, compute length of words, etc, maybe the format that is chosen is not best suited for the models. We can’t assume that the model well understands what “x elapsed tokens, y remaining” means. For example, a different ways of how to phare urgency to a model is:
--“x% time elapsed, y% remaining” - maybe the models have seen more training data in this format
- From the results, figure 6 looks like when using THOUGHTTERMINATOR the overthinking is reduced, but in some cases the performance is hurt. For example, in MATH500, the QwQ-32B (red dot), has pass@10 of ~0.9 without THOUGHTTERMINATOR, and with pass@10 of ~0.55. This suggests to me that a more thorough analysis of the failure cases when using THOUGHTTERMINATOR needs to be done. I would have appreciated a more systematic and thorough analysis of the failures, to make sure that the trade-offs of using THOUGHTTERMINATOR are well understood and outlined.

---

> ### Author Response · Authors · 2025-06-03
>
> 1. Prompt diversity and model’s ability to read tokens
>
> > Only one prompt format was tried; models may not understand ‘x elapsed tokens, y remaining’. Try alternative phrasings (e.g., ‘x % time elapsed’).
>
> Thank you for raising this question, we have added an additional urgency formats and re-run the key experiments on R1-1.5B (MATH500 & GPQA). The table below compares two interrupt message formats.
>
> **Dataset: MATH500**
>
> | Interrupt Message                                      | Acc. | Pass@5 | Pass@10 | Avg. Tokens |
> |--------------------------------------------------------|------|--------|---------|-------------|
> | "I have used x tokens, and I have y tokens left to answer." | 0.48 | 0.81   | 0.87    | 1589.94     |
> | "x% time elapsed, y% remaining"                        | 0.44 | 0.74   | 0.81    | 1586.42     |
>
> **Dataset: GPQA**
>
> | Interrupt Message                                      | Acc. | Pass@5 | Pass@10 | Avg. Tokens |
> |--------------------------------------------------------|------|--------|---------|-------------|
> | "I have used x tokens, and I have y tokens left to answer." | 0.21 | 0.55   | 0.75    | 1278.96     |
> | "x% time elapsed, y% remaining"                        | 0.23 | 0.63   | 0.76    | 1264.29     |
>
> The results show **comparable accuracy and token usage across both variants**.
>
> 2. Apparent accuracy drop for QwQ-32B (MATH500)
>
> Thank you for pointing this out. Upon reviewing our logs, we found that the low reported accuracy for QwQ-32B + ThoughtTerminator was due to a logging error. At submission time, only a subset of completions had finished running. The numerator (number of correct answers) was computed over this finished subset, while the denominator mistakenly included the full evaluation set.
>
> After submission, we have more time to re-run the complete evaluation, and we found that **QwQ-32B + ThoughtTerminator achieves Pass@10 = 0.80**, which is significantly higher than previously reported. We will update Figure 6 in the camera-ready version.
>
> 3. ZEBRA outlier in Figure 7
> > In figure 7, for the ZEBRA dataset, the base seems to use the most tokens but produce the worst (by a huge margin) result. This seems like an outlier...
>
> With the **original 5,000-token cap**, only a handful of the very hardest ZebraLogic items (difficulty > 7) were answered correctly, and those few successful traces happened to be concise.
> Because the plot averages only over correct answers, the higher difficulty bins were therefore represented by **a tiny set of short completions**, making it look as if “hard” questions required fewer tokens.
> After re-running **with a higher budget (max_tokens = 32,000)**, we obtained many more correct solutions, most of which are long, formal derivations.
>
> **The spend-versus-difficulty curve is now strictly increasing and the apparent outlier vanishes**. We will replace Fig. 7 with the updated version.
>
> 4. Readability of Section 2.1
>
> Thank you for the suggestion. We will revise §2.1 to improve clarity by adding inline explanations for key formulas and symbols, and adjusting the paragraph structure to ease readability.
>
> 5. Will the DUMB500 dataset be available publicly?
>
> Yes. We are currently packaging the dataset, grading scripts, and license files.

---

> > ### Comment · Reviewer_1isg · 2025-06-04
> >
> > Thank you for your answer. I will increase my score.

---

### Official Review · Reviewer_HZWD · 2025-05-12

**Rating:** 6
**Confidence:** 3
**Ethics Flag:** 1

**Summary:**

This paper proposed DUMB500, a dataset of extremely easy tasks to evaluate the calibration of reasoning models, and found that these models often overgenerate tokens, especially on easy problems. It also proposed ThoughtTerminator, a training-free decoding method that improves token efficiency by better calibrating reasoning depth to problem difficulty.

**Reasons To Accept:**

1. Good motivation to alleviate overthinking in reasoning model without retraining
2. The solution is very intuitive and straightforward.

**Reasons To Reject:**

1. The difficulty of a question is based on the (in)accuracy of model-generated answer, but the question difficulty is not the only factor that affects the answer accuracy, it can also relate to the question/answer memorization.
2. The task and chat evaluation in DUMB500 is evaluated by gpt-4o-mini, which is not a widely used model for LLM-as-a-judge. Would be nice to present the agreement between gpt-4o-mini and more widely used judge LLM such as GPT-4o to validate the evaluation effectiveness.
3. In figure 6, the performance of QwQ-32B + Terminator is significantly worse than other methods. Would be nice to have an analysis for that.
4. Would be nice to present more details on the prediction-based deadline estimator, including zero-shot gpt-4o-mini, and trained instruct-llama-8b model. Would be interesting to see how many data points can make the prediction lead to good performance.

---

> ### Author Response · Authors · 2025-06-03
>
> 1. On the difficulty definition and potential conflation with memorization
>
> In this work, we **do not claim to measure the intrinsic or absolute difficulty of a question**. Instead, we **define difficulty empirically**, based on the aggregated error rate across a diverse pool of models. This definition reflects how likely current models are to solve a problem correctly in practice, rather than its inherent complexity.
>
> > The question difficulty is not the only factor that affects the answer accuracy, it can also relate to the question/answer memorization.
>
> As clarified in Section 3, **DUMB500 was entirely human-authored from scratch**, and not sampled or modified from any existing benchmarks such as GSM8K, BBH, MATH, or HumanEval. The prompts are intentionally designed to be simple and novel, and do not follow typical benchmark phrasing or templates. Therefore, we did not consider memorization to be a major risk. Additionally, our difficulty scores are aggregated across a diverse pool of models, which further reduces the bias of any single model’s memorization behavior.
>
> ---
>
> 2. GPT-4o-mini as judge and its agreement with GPT-4o
>
> > Would be nice to present the agreement between gpt-4o-mini and more widely used judge LLM such as GPT-4o to validate the evaluation effectiveness.
>
> We thank the reviewer for suggesting a validation step. We evaluated the agreement between GPT-4o and GPT-4o-mini on DUMB500's task and chat subsets, and observed the following:
> - For DeepSeek-R1: agreement = 99.12%
> - For QwQ-32B: agreement = 91.15%
>
> These results indicate **strong alignment between the two judges**, especially for models with more stable output patterns. This supports our use of GPT-4o-mini as a lightweight but reliable approximation to GPT-4o for large-scale evaluation. We will report these agreement results in our manuscript.
>
> ---
>
> 3. Low accuracy of QwQ-32B + ThoughtTerminator in Figure 6
>
> Thank you for pointing this out. Upon reviewing our logs, we found that the low reported accuracy for QwQ-32B + ThoughtTerminator was due to a logging error. At submission time, only a subset of completions had finished running. The numerator (number of correct answers) was computed over this finished subset, while the denominator mistakenly included the full evaluation set.
> After submission, we had more time to re-run the complete evaluation, and we found that QwQ-32B + ThoughtTerminator achieves **Pass@10 = 0.80**, which is significantly higher than previously reported. We will update Figure 6 and the relevant discussion in the camera-ready version.
>
> ---
>
> 4. More details on the prediction-based deadline estimator
>
> Our deadline estimator predicts the difficulty of a question on a 1–10 scale and maps it to a token budget. We explored two variants:
> - A **zero-shot GPT-4o-mini baseline**, which predicts difficulty based on a fixed instruction-prompt format;
> - A **fine-tuned Instruct-LLaMA-8B**, trained on 1,465 examples covering diverse task types.
>
> Each sample consists of a question and a discrete difficulty label derived from the minimum correct generation length (discretized into bins). The trained predictor achieves a **mean absolute error (MAE) of 1.57** on a held-out test set, indicating that predicted difficulty levels are, on average, within 1–2 bins of the reference.
> We will include these training details, sample format, and a summary of the sub-sampling results in Appendix of the revised version.

---

> ### Author Response · Authors · 2025-06-09
>
> Hi Reviewer HZWD, we just wanted to check if you have any thoughts or concerns regarding our rebuttal. Thanks!

---

### Author Response · Authors · 2025-06-09
**Concluding response to all reviewers**

We’re happy to hear you find our work is **well-fitted to a gap in a timely problem**, containing **many interesting findings** which could **benefit and inspire the community** (aNwS). Reviewers HZWD and U1jN praise the **simplicity and effectiveness** of our proposed overthinking mitigation technique, ThoughtTerminator, and all four reviewers note the utility of DUMB500 in plugging a **new gap in difficulty evaluation**. Additionally, reviewers praised the **novelty of our method for difficulty measurement** and praised the elegant simplicity of all parts of our paper.

A common complaint reviewers raised was the **number of baseline decoding methods**. Thankfully, you suggested several that were straightforward to implement. To our excitement, all three of the proposed baselines both strengthen our demonstration of ThoughtTerminator's performance, and unlock useful insights into both why it works and the nature of reasoning models more broadly.

1. Rephrasing the interrupt messages from token count to % of tokens remaining (1isg)
2. Simply prompting the model to not overthink/answer concisely (aNwS)
3. Tell the model how many tokens have been spent without saying how many remain, and not applying constrained decoding (U1jN)

These are all great and intuitive ablations to try, and we ran all of them on R1-1.5B and QwQ-32B. We found that **ThoughtTerminator continues to outperform all of them, but they reveal interesting additional insights**. (1) demonstrates that *varying the token message has only a modest impact on performance*, (2) demonstrates that *the model ignores simple telling it to not overthink*, and (3) demonstrates that Terminator’s *forced answering with constrained decoding is necessary* for the model to benefit from interrupt messages. Furthermore, (2) and (3) further support the general hypothesis that reasoning models are not really “aware” of how many tokens have been spent in an actionable manner, and require some external control over decoding to enforce token budgets. We are excited to add all of these results and findings to the camera ready, run on all models rather than just r1-1.5B and QwQ-32B.

Some reviewers found the unconventional structure of the paper or some of the terminology confusing. We apologize for any lack of clarity in the paper, and believe we have adequately answered your concerns in our responses. All of the clarifications provided in the rebuttal will be integrated into our camera ready revision.

We are very grateful to all for your insightful reviews and look forward to hearing your thoughts regarding our rebuttals. We appreciate how your suggestions have allowed us to strengthen our work, and are happy to discuss any other lingering thoughts before the end of the discussion period.

---

### Decision · Program_Chairs · 2025-07-08

**Decision:**

Accept

**Comment:**

This paper focuses on over-thinking, the problem of reasoning models generating many more tokens than necessary.  The paper contributes both a manually created dataset of easy questions for measuring overthinking, as well as an inference-time technique for reducing overthinking.

Strengths:
- The problem studied is timely and impactful
- The manually created dataset is distinct from existing resources and can be useful for others working on over-thinking
- The interrupt based solution is effective - often not reducing accuracy and at the same time reducing the number of tokens by about 2x over multiple baseline models

Weaknesses:
- The original manuscript was missing some natural baselines and comparisons.  However during the author response period, the authors provided the baselines requested by the reviewers, and the results are all still in favor of the proposed technique.